# An Evaluation of the Drone Delivery of Adrenaline Auto-Injectors for Anaphylaxis: Pharmacists' Perceptions, Acceptance, and Concerns

**September Beck [1], Tam T. Bui [2], Andrew Davies [3], Patrick Courtney [4], Alex Brown [5], Jef Geudens [5] and Paul G. Royall [1,\*]**

1   Department of Pharmacy, Faculty of Life Sciences & Medicine, Institute of Pharmaceutical Science, King's College London, 150 Stamford Street, London SE1 9NH, UK; september.beck@kcl.ac.uk
2   Biomolecular Spectroscopy Centre, Optical and Chiroptical Spectroscopy Facility, Kings College London, London SE 11UL, UK; tam.bui@kcl.ac.uk
3   Hospital Pharmacy & Medicines Optimisation Team, NHS England London, Skipton House, 80 London Road, London SE1 6LH, UK; andrew.davies@nhs.net
4   Tec-connection, Oberlohnstr 3, D78467 Konstanz, Germany; patrick.courtney@tec-connection.com
5   Skyports, Unit LG.02, Edinburgh House, 170 Kennington Lane, London SE11 5DP, UK; alex@skyports.net (A.B.); jef@skyports.net (J.G.)
\*   Correspondence: paul.royall@kcl.ac.uk

**Abstract:** Anaphylaxis is a life-threatening condition where delays in medical treatment can be fatal. Such situations would benefit from the drone delivery of an adrenaline auto-injector such as EpiPen®. This study evaluates the potential risk, reward, and impact of drone transportation on the stability of adrenaline during episodes of anaphylaxis. Further, this study examines pharmacists' perceptions on drone delivery—pharmacists approved the use of drones to deliver EpiPen® during emergencies but had concerns with drone safety and supply chain security. Laboratory simulated onboard drone conditions reflected typical missions. In these experiments, in vitro model and pharmaceutical equivalent formulations were subjected independently to 30 min vibrations at 5, 8.43, and 13.33 Hz, and temperature storage at 4, 25, 40, and 65 °C for 0, 0.5, 3, and 24 h. The chiral composition (an indicator of chemical purity that relates to molecular structure) and concentration of these adrenaline formulations were determined using ultraviolet (UV) and circular dichroism spectroscopy (CD). Adrenaline intrinsic stability was also explored by edge-of-failure experimentation to signpost the uppermost limits for safe transportation. During drone flight with EpiPen®, the temperature and vibration g-force were 10.7 °C and 1.8 g, respectively. No adverse impact on adrenaline was observed during drone flight and laboratory-simulated conditions shown by conformation to the British Pharmacopeia standards ($p > 0.05$ for CD and UV). This study showed that drone delivery of EpiPen® is feasible. There are more than 15,000 community pharmacies and ≈9000 GP surgeries spanning the UK, which are likely to provide achievable ranges and distances for the direct drone delivery of EpiPen®. The authors recommend that when designing future missions, in addition to medicine stability testing that models the stresses imposed by drone flight, one must conduct a perceptions survey on the relevant group of medical professionals, because their insights, acceptance, and concerns are extremely valuable for the design and evaluation of the mission.

**Keywords:** adrenaline; EpiPen®; stability; chirality; drone delivery; pharmacist; perception

## 1. Introduction

Uncrewed aircraft vehicles (UAV) were originally designed for the military to use for surveillance, to serve as weapon platforms, and to simulate conventional aeroplanes for target practice [1]. However, technological advancements have refined the UAV industry, taking flight to a new era, whereby design innovation, widespread production, and even miniaturisation have captured the imagination of the civilian sector [2]. Today, these vehicles are more commonly known as drones. A drone is defined as an uncrewed aircraft vehicle typically flown by a ground controller [2].

Drones are now considered to be "multi-tools" that bring extraordinary value across diverse applications [3–11]. These niche robots have a unique and flexible platform, and they are capable of transforming the airspace and creating a market for urban air mobility. Drones are currently employed by marketing agencies and digital technology companies; they are used in cinematography [3], rescue operations [4], monitoring weather patterns, agricultural expansion [5], deforestation [6], and delivering packages [7]. As we enter the robotic era, opportunities for drones and robotic systems are growing exponentially [8]. US investment into drone technology will reach $506 million in 2019 and likely to create $82 billion in commercial activities during the next ten years [9]. However, the increased universal popularity of drones has led to episodes of market disturbance and dangerous events such as the chaotic incident at Gatwick Airport on 19th December 2018. A consequence of this disruption has been the implementation of the "UK Drone Code" by the Civil Aviation Authority (CAA) and the enhancement of UK regulations.

Drone literature identifies the prospect of a viable healthcare service driven by drones with endless possibilities for both medical staff and patients [6]. Drone utilisation has sparked the potential to act as a professional tool in transforming healthcare and patient outcomes. The National Health Service, NHS, has recently announced their involvement in the UK's first UAV pilot project [10]. The first successful case of drone use, the "Uber for Blood" service [8], was established by Zipline and the United Parcel Service, who delivered blood to over 20 remote regions in Rwanda, Africa where there is a shortage of blood types [7]. Transporting organs such as kidneys [11] has sparked excitement in the medical world, whereby companies such as EHang have started to drone deliver transplant organs in less than 30 min, avoiding the delays of conventional transport. Matternet has successfully delivered medicines to Haiti after the earthquakes [12], and drones have also successfully delivered defibrillators for emergency cardiac care [13]. These pilot studies show that drone delivery is feasible but provide little consideration of the medicine stability and quality that must be assessed to maintain patient safety.

Urban air mobility is an attractive platform with the ability to increase the accessibility and on-demand availability of medicines [8]. Drone delivery can overcome the logistical challenges experienced by conventional transport methods. Drones also have the capability to reach and navigate rural, remote, and inaccessible areas. A reduction in transit time [14] means faster medication delivery and the possibility of saving more lives where time is of the essence. There is also a likely reduction in carbon dioxide emissions generated by this route of transport and in the cost of operation when installed and used frequently [15].

This study focusses on the emergency requirement for anaphylaxis. Such situations would benefit significantly from the utilisation of drone delivery, where the risk–reward ratio is low and where conventional transport has failed to deliver an expeditious emergency service [16].

Anaphylaxis is a rapid and serious systemic allergic reaction with a lifetime prevalence of 2% [17], which is commonly triggered by food [18], medications, or insect stings [19]. A wide range of symptoms including shortness of breath and hives appear within 5–10 min of the trigger [16–19] and gradually worsen. Anaphylaxis can become deadly in less than 15 min without sufficient treatment. There has been an increase in child anaphylaxis (72%) and adult anaphylaxis (34%) hospital admissions over the past 5 years in England, with the largest increase in admissions (167%) occurring in London [20]. Adrenaline (also known as epinephrine) is the only first-line medication for anaphylaxis treatment [21] where prompt injection is crucial to preserve a patient's life. Adrenaline is licensed as a dilute

solution for injection in the forms of ampoules, auto-injectors, pre-filled syringes, and inhalers [22], with patient preference residing in the auto-injector subtype [23]. This study focuses on EpiPen® 1mg/mL adrenaline auto-injectors.

NHS clinical guidelines recommend that those with potentially serious allergies should carry two in-date adrenaline auto-injectors as an emergency treatment for anaphylaxis. Patients who have allergies towards insect stings, peanuts, milk, and seafood fall within this group; furthermore, people with asthma and the allergic skin condition atopic eczema have a greater risk of developing anaphylaxis. The National Institute for health and Care Excellence (NICE) guidance states that it is important to use an adrenaline auto-injector as soon as possible if an anaphylactic reaction is suspected.

Regardless of episode severity/episode uncertainty, patients are encouraged to administer adrenaline as soon as their symptoms appear [24]. Prompt administration produces positive patient outcomes whilst reducing the possibility of the need to administer multiple doses. However, up to 35% of patients require a second dose of adrenaline, which can be administered 5–15 min after the initial injection (usually a result of recurring symptoms) [24]. Adrenaline mean time to peak plasma concentration is $8 \pm 2$ min after injection, with effects wearing off after 10–20 min [25].

Community pharmacists are particularly well positioned to respond to medical emergencies, which is a result of extensive facility distribution patterns; yet, they are underutilised. Thus, the drone delivery of adrenaline auto-injectors directly from community pharmacies would be a highly beneficial solution and convenient in an anaphylactic emergency. Within community settings, pharmacists are perceived as responsible, trustworthy individuals in regard to medicines administration and healthcare advice on the frontline. Furthermore, pharmacists have a unique role in issuing emergency prescriptions, which is extremely advantageous for a future drone delivery service—i.e., on demand availability and accessibility of adrenaline auto-injectors. It is crucial to conduct a survey on pharmacists' perceptions of drone delivery, as their insights will be valuable in shaping the proposed missions.

Patients who have a known risk of anaphylaxis should carry at least one auto-injector [21] with them due to the danger of an unpredictable onset, with the UK government recommending the carrying of two auto-injectors [18]. Manufacturers recommend that, "Adrenaline should be stored in a dark place, protected from light, at room temperature 20–25 °C (where excursions 15–30 °C are permitted). Adrenaline itself should not be stored in the refrigerator" [26]. Storage outside these conditions causes failure in optimal dose ejection, problems of degradation [27], and a decrease in efficacy [28]. Any compromise in adrenaline efficacy can be the difference between life and death. However, in the real world, strict adherence to these storage conditions for individuals may not always be possible and may conflict with the idea of ensuring a reliable dose [29]. There are currently universal supply chain issues [30] with EpiPen®—they are becoming increasingly unavailable and unaffordable [21] as manufacturers are increasing their prices, which impacts greatly on patient safety [31].

Adrenaline chirality (a geometric property based on molecular symmetry) and instability has been confirmed by multiple studies [32]. However, only a minority address the importance of changes in chiral composition within formulations such as EpiPen® [33]. As a chiral molecule, adrenaline has the capacity to degrade and racemise from its bioactive isomer (−)-adrenaline to its bio-inactive isomer (+)-adrenaline [34]. As a result, this causes a decrease in Active Pharmaceutical Ingredient (API) potency [28] within formulations and a reduced health outcome; thus, it is vital to maintain the dose uniformity with respect to (−)-adrenaline. Circular dichroism spectroscopy [35] (CD) has been used in this study to simultaneously analyse the chiral composition, concentration, and enantiomeric stability of adrenaline formulations. The presence of (+)-adrenaline and changes in adrenaline concentration outside of the British Pharmacopeia [36] (BP) label claim (%) have been selected as the indicators of instability.

The goods classifications for pure adrenaline (drug substance) and when formulated into an injection for an EpiPen® (drug product) are quite different in terms of the transport perspective. Adrenaline (epinephrine) in its solid, acid tartrate form, is classified, according to the British Pharmacopoeia safety data sheet (1907/2006/EC, Article 31), as a toxic solid, UN2811, packing group III

(presenting low danger), and it has an International Air Transport Associated (IATA) classification of 6.1. This means that the transportation of pure adrenaline by air requires performance-oriented packaging that must have enough strength to withstand shocks, loadings, and the typical changes in atmospheric pressure that occur during flight. This would normally be achieved by a primary container that is usually plastic—for example high-density polyethylene, which is inert and able to avoid leakage at an internal pressure of 95 kPa. The primary package would be placed within rigid outer packaging but separated by a cushioning material, e.g., polystyrene or engineered cardboard. A clear label would be required, following UN guidelines indicating toxic material and the IATA hazard class of 6.1. Transport of 5 kg is permissible on a passenger aircraft and 50 kg is the limit for transportation on a cargo aircraft. Many medicines, especially EpiPen® adrenaline auto-injectors, are not treated as dangerous goods. Medicines that are dangerous for carriage are those that contain flammable or volatile ingredients—for example, the propellants used in a pressurised metered dose inhalers—or those that are particularly toxic, for example, chemotherapies and the cytotoxic drugs used in the treatment of cancer; these are covered by UN numbers 1851, 3248, and 3249. These numbers are used to identify hazardous substances within international transport systems and are assigned by the United Nations Committee of Experts on the Transport of Dangerous Goods.

The rationale for not subjecting medicines to IATA transport regulations is based on the small amounts of the active pharmaceutical ingredient (or drug) present, typically in the milligram range, which is far below the UK's Health and Safety Executive's limited quantity exemption for the international carriage of dangerous goods, and the acknowledgement that the medicine is contained within packaging already approved for retail sale and distribution for personal and household use (UN-SP601). In the UK and Europe, packaging approval is governed by the Medicines and Healthcare products Regulatory Agency and the European Medicines Agency respectively, within their Good Distribution Practice guidance. Similar regulations are applied worldwide, e.g. by the Food & Drug Administration, FDA, in the US. Thus, solid adrenaline falls within the regulations concerning the air transportation of dangerous goods, but adrenaline auto-injectors do not. For example, the materials safety data sheet for adrenaline (epinephrine) injection (1 mg/mL) published by a manufacturer of the drug, Hospira & Pfizer, states that the injection is not regulated for transport under USDOT, EUADR, IATA, or IMDG regulations. (US Department of Transport, USDOT; European Agreement concerning the International Carriage of Dangerous Goods by Road, EUADR; International Maritime Dangerous Goods code, IMDG). Therefore, the authors' choice of EpiPen® adrenaline auto-injectors allows the evaluation of drone delivery without the requirement for extra packaging, and it keeps the studies' focus on the key research questions, i.e., the impact of drone flight on adrenaline stability and the perceptions of pharmacists towards drone deliveries. It should be noted that CAA approval is required for all commercial operations, and medical deliveries are no exception. Many medical cargos of interest are classified as dangerous goods—for example, samples used in blood tests (covered by UN3373); thus, robust safety cases will be required to gain regulatory approval for all future UAV commercial medical deliveries.

EpiPen® auto-injectors have successfully passed drop tests and load challenge, without any loss of performance characteristics; i.e., after the mechanical stress, EpiPen® was able to consistently deliver the correct volume dose, but the chemical stability of adrenaline was not measured [37]. Mechanical robustness is not surprising, as EpiPens® are designed in rigid packaging to allow them to be carried in bags and coats with limited risk. Therefore, the specific safety perspectives associated with transport of EpiPens® by drone are adrenaline stability during flight and unexpected or unplanned landing. In the case of unplanned landing, tampering and diversion need to be considered, as unlike road carriage, no operator would be in the immediate vicinity to protect the cargo. The outside of EpiPen® packaging has clear safety warnings, and there is little financial benefit in the UK associated with diversion (unregulated re-sale) because of the national prescription prepayment scheme. This may not be the case internationally; for example, EpiPen® costs approximately $400 in the US, which would warrant the development of extra protection. Anti-tampering measures would also reduce the risk of

bystanders harming themselves if they are unfamiliar with EpiPen®; such themes are taken up at the end of this paper.

To date, there are no published works on the transportation of adrenaline auto-injectors by drone. Therefore, the aim of this study was to investigate the potential benefits, risks, and impact of drone transportation on the stability of adrenaline. This investigation will also examine pharmacists' perceptions of drone delivery of EpiPen®, test the intrinsic stability of adrenaline by exploring the edge-of-failure, and expose various adrenaline formulations to a range of simulated onboard drone conditions and drone flight.

## 2. Materials and Methods

This study embraced a systematic quality by design approach with methodologies expanding from conventional International Council for Harmonisation [38] (ICH) stress tests (Table 1) and BP criteria (Table 2). All equipment was calibrated according to ICH guidelines, with sample preparation and storage optimised to ensure Active Pharmaceutical Ingredient (API) stability and performance.

**Table 1.** Critical material attributes and process parameters considered for EpiPen® drone flight and laboratory simulation. Critical material attributes are a set of properties for an input material within a given range to ensure the desired material output quality. Critical process parameters are factors that must be controlled and/or monitored due to their ability to impact on critical quality attributes of the medication that ultimately affect product quality, safety, and efficacy.

|  | **Parameter** | **Measurement and Method** | | |
| --- | --- | --- | --- | --- |
| **Critical Material Attribute** | Chiral Composition and Purity | Circular Dichroism | | |
|  | Enantiomeric Concentration | UV-Vis Spectroscopy | | |
|  | Visual Appearance | Visual Inspection | | |
|  | **Parameter** | **Conditions** | **Time Course** | **Test Scenario** |
| **Critical Process Parameter** | Temperature | 4–65 °C | 0, 0.5, 3, 24 h | Lab Stress Test |
|  | Vibration | 0–13.33 HZ | 0, 0.5, 3, 24 h | Lab Stress Test |
|  | Natural and UV Light | Natural light and UV | 0, 0.5, 3, 24 h | Lab Stress Test |
|  | Duration, Pressure, and %Relative Humidity | Ambient | 18 min | Drone Flight |

**Table 2.** British Pharmacopeia (BP) criteria [36] for adrenaline injection/epinephrine injection. Dose uniformity and concentration calculated using $\varepsilon 280 = 2700$ M$^{-1}$ cm$^{-1}$ and Mw = 183.2 g/mol [28,39].

| | **British Pharmacopeia Criteria:** |
| --- | --- |
| Colour and Clarity Test | Physical stability of all solutions was determined by visual inspection against a black and/or white background using standard laboratory lighting. Solutions were examined for signs of colour change, precipitate formation, or particulate matter. |
| Concentration and Dose Uniformity | Concentrations were calculated using absorbance readings and the Beer Lambert Law, $\varepsilon 280 = 2700$ M$^{-1}$ cm$^{-1}$ and Mw = 183.2 g/mol. These were compared to the BP criteria, which states that adrenaline injection must contain 0.090 to 0.110% *w/v* and not less than 0.085% *w/v* is (−)-adrenaline. |
| pH Test | pH must be maintained between 2.8 and 4.0. |

### 2.1. Pharmacists' Perspectives Survey

Fifty-five pharmacists from a broad range of expertise completed a survey that examined their perceptions on the drone delivery of EpiPen® to patients in emergency situations. This questionnaire was posted in the NHS Improvement Pharmacy and Medicines Optimisation Newsletter to Trust Chief Pharmacists on 25 November 2019. The full list of questions and the participant information sheet (which were available as a hard copy and URL link) can be found in the Supplementary Information (Supplementary SI 1).

### 2.2. Exploration of Stability Forced Degradation

Serial dilution of (−)-adrenaline (LKT Laboratories, Inc St. Paul, MN, USA, >99.5%) in 0.1 M HCl and in distilled water were prepared from a 0.5 mg/mL stock and exposed to light using an artificial lamp with visible and UV outputs and natural laboratory light at 25 °C, over a 24 h period with CD analysis and visual inspection at regular intervals. This was repeated using (−)-adrenaline (+)-bitartrate salt (Sigma Aldrich, Gillingham UK, E4375-1G, >98%) and adrenochrome (Sigma Aldrich A5752-25MG, >95%). This formed a set of control samples that were known to have suffered degradation.

### 2.3. Preparation of Buffer and Model Formulations

Model formulations were created due to difficulty in sourcing a large supply of EpiPen® and the API, adrenaline, in its acid tartrate form.

In vitro (−)-adrenaline and (−)-adrenaline (+)-bitartrate model formulations and a buffer solution (SI 2) were prepared based upon the EpiPen® formulation. A second buffer solution (Supplementary SI 2) was also prepared following the injection ampoule formulation (Macarthys Laboratories Ltd, Romford, UK, T/A Martindale Pharma PL01883/6118R). Fifty mg (−)-adrenaline were weighed in a dark room and added to 100 mL buffer solution prior to experimentation. Unless otherwise stated, the concentration of (−)-adrenaline, (−)-adrenaline, (+)-bitartrate, and the ampoules were maintained at 0.5 mg/mL, which are optimised conditions for CD. All samples were prepared immediately before experimentation and vials were wrapped in silver foil due to (−)-adrenaline photosensitivity.

### 2.4. Stress Tests

#### 2.4.1. Vibration

A vortex mixer (ZX3, VELP) fabricated the range of vibrations that the EpiPen® would experience during drone take-off, flight, and landing. Glass vials (7 mL) filled with 2 mL of in vitro model solution (0.5 mg/mL) were fixed onto the mixer using tape and subjected to vibration for 30 min at 25 °C: 300 rpm (5 Hz), 500 rpm (8.43 Hz), and 800 rpm (13.33 Hz). A control sample (0 rpm, 0 Hz) was prepared for comparison. Hii et al. correlated the vortex mixer speed to vibrational frequency using vibration data loggers and mapped these using the same loggers to the vibration and acceleration g-forces experienced during multirotor drone flight [40]. During vortex mixing, an off-centered rubber cup oscillates in a circular motion, the sample vial is placed into this cup, and the oscillatory motion is transmitted to the liquid inside, creating a vortex. This intense shaking and turbulent flow induce shear forces and cavitation within the solution. Typical rotational speeds for the propellers of small multirotor drones are between 4000 and 6000 rpm, and they are typically near 5000 rpm in flight [41]. This is much greater than the vortex mixer can achieve. However, the symmetrical alignment of rotors in such drones is designed to reduce vibration and only in lift-off and take-off are significant vibrations observed; for Hii et al., this was 3.4 Hz, with an upper limit peak of 10 Hz. During normal horizontal flight, this dropped to only 0.1 Hz. However, when rotors are damaged or poorly balanced, vibration will increase with a concomitant lowering of thrust performance [42]. The g-force also needs to be considered, so in the present study, this was measured directly on board the flight, see Section 2.5 and Figure 1d, and the stability of the transported adrenaline measured post flight, after exposure to this force.

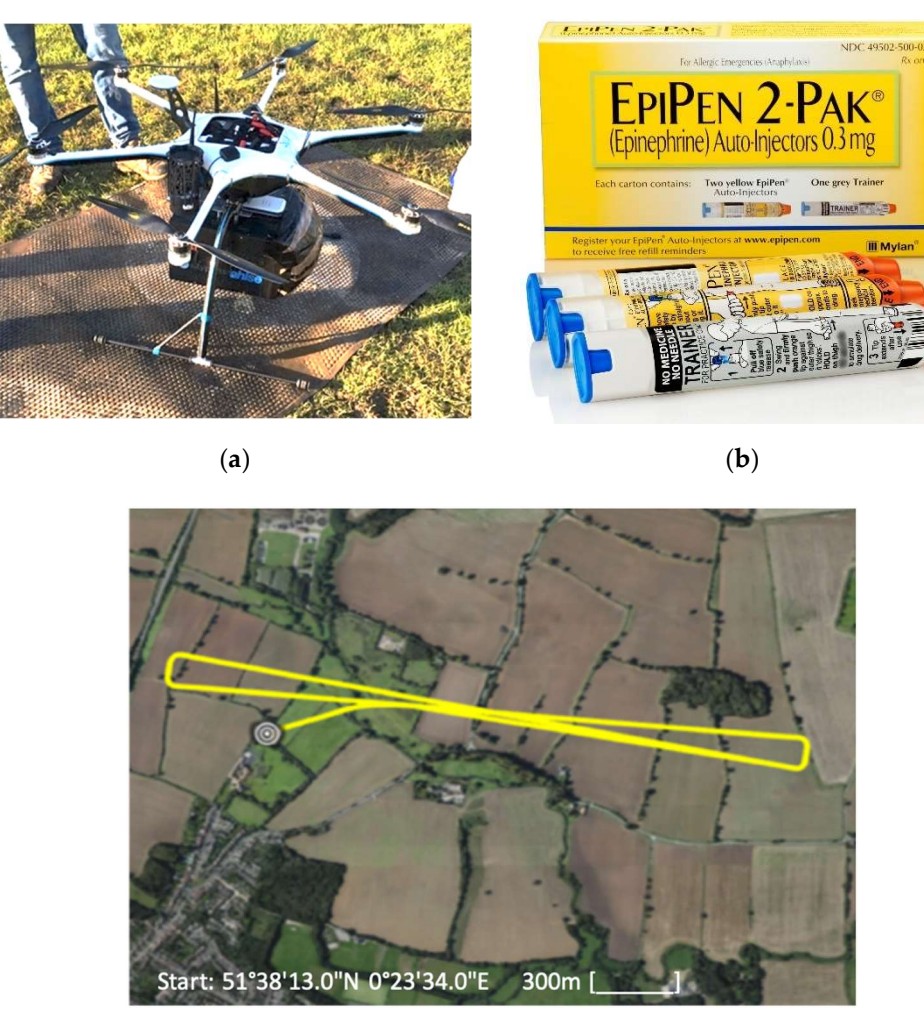

(a)                                                                   (b)

(c)

Flight Specifications:

| | | | |
|---|---|---|---|
| **Distance Flown** | 10732m | **Vibration g-force** | 1.82g (± 0.23) |
| **Cruise Speed** | 12m/s | **Temperature** | 10.70°C (± 0.32) |
| **Max Speed** | 12.4m/s | **Humidity** | 47.74%rh (± 0.29) |
| **Altitude AGL** | 61m | **Dewpoint** | 0.06°C (± 0.35) |
| **Altitude AMSL** | 116m | **Windspeed** | 5.92mph |

(d)

**Figure 1.** (**a**) Clogworks Dark Matter HX Drone at Skyports' Billericay Flight Site, CM12 0ES. Drone maximum take-off weight = 20 kg and unloaded mass = 5.30 kg. The ultra-low vibrations, extensive operating temperatures (between −20 and 45 °C), excellent chassis strength, wide payload range options, and its aerodynamic, light-weight design enhance performance. (**b**) Image of EpiPen® used in this investigation: Adrenaline auto-injectors 0.3mg Mylan Products Ltd. UK, POM [39,43]. (**c**) Billericay, Essex Flight Map. Image taken from Skyports SmartAp and controllers on 04/12/19. The mission highlighted in yellow followed Civil Aviation Authority (CAA) protocols with a 35% battery minimum required for safe landing. (**d**) Flight Specifications. The table provides the calculated means and standard deviations for the mission. AGL = above ground level, AMSL = average mean sea level.

### 2.4.2. Temperature

Glass vials (7 mL) containing 2 mL in vitro model solution (0.5 mg/mL) were incubated at 4 °C (fridge), 25 °C (room temperature), 40 °C (incubator), and 65 °C (water bath) for 0 h (control), 0.5 h, 3 h, and 24 h respectively prior to UV and CD measurements. To ensure temperature consistency, an empty cuvette was incubated in the instrument cell holder thermostat (Quantum Northwest, TC125) at the required temperature; then, the solution was transferred, and the measurement was recorded. The temperature was measured directly with a 0.25 mm thermocouple probe in the sample solution. EpiPen® adrenaline auto-injector manufacturers and the MHRA recommend storage and transport at temperatures are between 20 and 25 °C, but temperature excursions between 15 and 30 °C are permitted.

### 2.4.3. Circular Dichroism Spectroscopy

The CD and UV spectra were recorded in triplicate on the APL Chirascan spectrophotometer after samples were subjected to simulated and flight conditions, with the following parameters: 700–200 nm Wavelength, 2 nm Bandwidth, 1 nm step-size, 0.5 s accumulation time-per-point. Ten, 2, and 0.5 mm Quartz Suprasil rectangular cuvettes (Starna Scientific Ltd, Hainault UK.) were used to accommodate the different concentrations. The instrument was flushed with pure evaporated $N_2$ continuously throughout the experiment. UV and CD spectra were subtracted with their appropriate buffer spectrum and smoothed with a window factor of 8 using Savitzky–Golay Filter for better presentation (Pro-Data Viewer v4.2.18).

### 2.5. Drone Flight

A Clogworks Dark Matter HX, carbon fibre drone (1565 × 1432 × 650 mm) (Figure 1a) powered by two 2.5 kg Tattu 22,000 mAh 22.2V LiPo Batteries was used for flight and controlled using a SmartAp, which was commissioned via Skyports (SI 3). This flight modelled a potential trip in an urban environment. Six EpiPen® (3 boxes: 2× Adrenaline Auto-injectors 0.3 mg Mylan Products Ltd. UK, POM, PL 46302/0171, LOT:9KB252P, Exp:11/2020) were brought to a Skyports Billericay flight site (Figure 1c) in their original carboard packaging (Figure 1b). In this study, the same batch and lot number of EpiPen® were used to minimise any manufacturing concentration discrepancies.

Four EpiPens® in their original packaging and the buffer solution for dilution within a protective bag (mass = 620.03 g) were sealed into the "temporary transport device" drone compartment (total mass of drone and load = 1.70 kg) using tape and flown in the line of sight for 18 min. One EpiPen® acted as a control.

Flight protocols implemented by the CAA were followed. Specifically, the drone flights were conducted under Skyports' permission for operations and strictly adhering to their approved safety protocols at all times. Skyports is approved by the Civil Aviation Authority to operate within visual line of sight (VLOS) and within extended visual line of sight (EVLOS) anywhere in the UK. Additionally, Skyports has permission to operate Beyond Visual Line of Sight (BVLOS) in certain locations. The flights at Billericay were conducted under Skyports EVLOS regulatory approvals. This permission granted by the CAA includes a full Operating Safety Case (OSC) prepared by Skyports, which extensively details the operation, processes, safety procedures, and emergency procedures that comprise a drone delivery operation.

Thermograms were taken using a DJI Mavic 2 Enterprise Dual Drone—four safety propellers were removed and images were taken by holding the drone, using conventional and thermal cameras. The temperature, humidity, and vibration frequency during the flight were monitored every 15 s. The temperature, humidity, and dew point were measured using the EasyLog LASCAR EL-USB-LCD and software v.7.6. Vibrations were measured using EXTECH VB300 3-Axis G-force USB and software v3.0.1.

After flight, the EpiPens® were transported back to the laboratory and diluted to 0.5 mg/mL for CD analysis.

*2.6. Statistical Analysis*

Two-tailed unpaired t-tests were conducted assuming equal variance, using Microsoft Excel 2019 to determine any statistically significant differences. Results were considered statistically significant where $p < 0.05$.

## 3. Results and Discussion

*3.1. Stability Investigation*

Forced degradation and edge-of-failure protocols were conducted following ICH Q1A (R2) guidelines [38] to confirm the intrinsic stability of (−)-adrenaline and its degradation pathway. This exploration focused specifically on changes in (−)-adrenaline chiral composition and concentration in different aqueous injection solutions using CD and UV, respectively (Supplementary SI 4). ICH represents the International Council for Harmonisation of the technical requirements for pharmaceuticals for human use, and Q1A (R2) guidelines are associated with stability testing requirements. The purpose of such stability testing is to provide evidence on how the quality of a drug or medicine varies with time under the influence of environmental factors, for example temperature and humidity. For EpiPen® adrenaline auto-injectors, such testing has already been carried out, and quality indicating analytical procedures has allowed the recommend storage conditions and shelf life to be set. The approach here was to use UV (ultraviolet) and CD spectroscopy assays to detect changes in concentration and chiral composition resulting from a loss of stability and the generation of well-documented degradation products; see Figures 2 and 3. Thus, quite simply if the adrenaline solution changed colour or became cloudy, or the UV or CD spectra differed from the control; then, it would be concluded that it was unstable towards the particular environmental challenge investigated. Likewise, the absence of colour change and no significance difference in both the UV and CD spectra indicated stability within the experimental time frame. EpiPens are enclosed units, so humidity was not considered, but temperature and vibration were considered as environmental factors. Vibration originating from drone flight has been addressed in Section 2.4.1. The increase in load mass reduces the range of a drone flight [40], so here, the flight stages were modelled without insulation or temperature control, as both would add mass. The World Health Organization, WHO, has designated four climatic zones—temperate, subtropical, hot and dry, and hot and humid. The maximum accelerated stability testing temperature representing hot climates is 40 °C, the highest recorded temperatures are in the low 50s; thus, to explore the edge of failure, a 65 °C temperature limit was investigated here. Both WHO and the ICH recommend stability testing at normal refrigerator temperatures, and so 4 °C was chosen as the lower limit. Our study has been designed to conform to worldwide aviation regulations, and so flight altitudes would be within 400 ft and ground temperatures may be extrapolated to this relatively low height. Drone operations are occurring in all WHO-designated climatic zones; thus, the stability environments investigated are well within the temperatures potentially encountered during loading, lift-off, flight, and the landing stages of future drone operations.

Adrenaline is an inherently unstable catecholamine [28]. It has been reported that adrenaline's chirality and amino moiety trigger sensitivity to degradation [32] by oxidation (Figures 2a and 3), autooxidation, racemisation, and excipient interaction [28]. Oxidation and racemisation are known to proliferate in alkaline solutions [44], upon exposure to light and temperature changes [45], but they become retarded under acidic conditions, as confirmed by Figure 3.

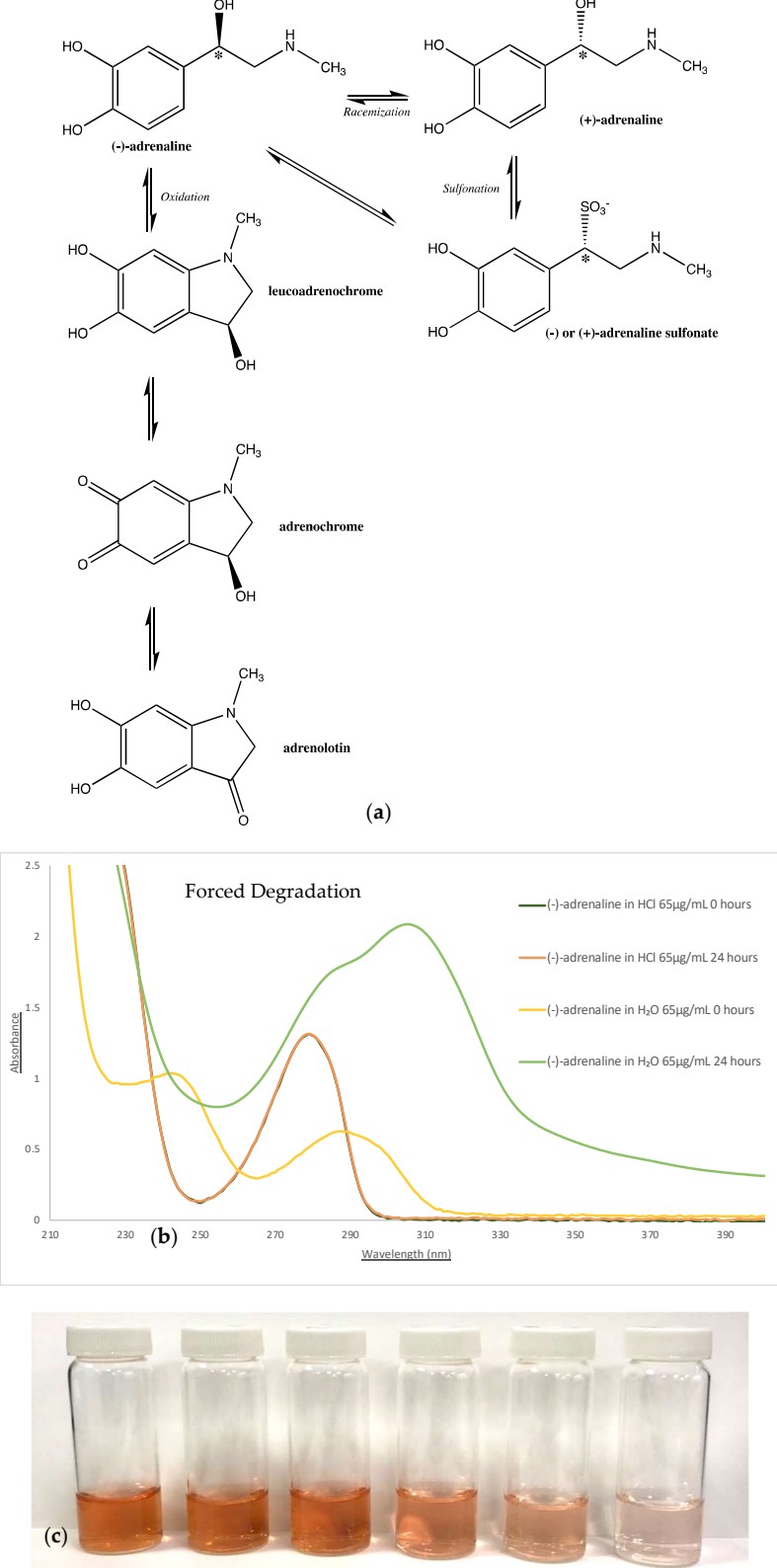

**Figure 2.** (**a**) The multi-step reversible pathway of (−)-adrenaline (L-adrenaline) degradation. (**b**) UV spectra of (−)-adrenaline in 0.1 M HCl and water at 65 µg/mL for 0 and 24 h. The design of the circular dichroism spectroscopy (CD) instrument and the cuvette path length permitted absorbances >1 to be calculated. (**c**) Visual appearances: Serial dilution of (-)-adrenaline in water after a 24-h period at 25 °C. The red colour intensity increases as the concentration increases L → R 65, 50, 40, 20, 11, and 10 µg/mL. Picture taken against a white background.

**Figure 3.** Cyclisation of (−)-adrenaline (L-adrenaline) into its degradation products.

Under acidic conditions, the UV spectra of adrenaline revealed distinctive peaks at 280 nm and 221 nm, and the CD spectra exhibited a negative ellipticity signal at 230 nm. These arise from the phenol chromophore electronic transitions $\pi \to \pi^*$, which are indicative of (−)-adrenaline [44]. After 24 h of exposure, there were no significant differences in UV ($p = 0.96$) and CD ($p = 0.23$) when compared to the control and no visible changes; the solutions remained clear and colourless with no precipitation or turbidity (Supplementary SI 4).

However, after exposure to alkaline conditions for 24 h, the UV spectra of adrenaline exhibited a bathochromic and hyperchromic shift in $\lambda_{max}$ from 280 nm to 300 nm. This shift is indicative of the adrenaline multistate transitions, where adrenaline oxidises into various degradative o-quinone structures arising from the charge transfer between the benzene chromophore and oxygen [44] (Figure 2b).

After 24 h, the CD spectra revealed a strong positive ellipticity signal at 230 nm, confirming a switch in the rotation of the chiral centre and the presence of degradant (+)-adrenaline. The positive and negative peaks at 300 nm are independent transitions as adrenaline oxidises, which are reflective of the magnetic dipole moment around the benzene chromophore and the carbonyl electron transfer. Furthermore, these solutions distinctly changed colour at different intensities depending on their concentration over 24 h. Solutions changed from colourless to pink and intensified to red as more freely oxidisable compounds formed—but neither turbidity nor precipitates were observed. There were significant differences ($p < 0.05$) in both UV ($p = 1.26 \times 10^{-8}$) and CD ($p = 5.47 \times 10^{-16}$) after 24 h when compared to time zero.

Dehydrogenation and cyclisation are the initial oxidation processes producing o-quinone structures that can spontaneously rearrange in solution [28]. The consecutive rapid oxidation and auto-oxidation of these structures subsequently forms coloured yet pharmacologically inactive indole derivatives [32]. The major indole-derivative degradant adrenochrome sets up a redox cycling process advancing oxidation and auto-oxidation [44]. Under acidic conditions, cyclisation is prevented.

Adrenochrome $\lambda_{max}$ is reported at $\approx$310 nm and 480 nm [46] with a minima 360 nm, resulting from the aniline chromophore and neighbouring interacting $\pi \to \pi^*$ transitions. It was difficult to distinguish whether the (−)-adrenaline in water solutions after 24 h contained adrenochrome, as we were unable to view discrete peaks at these wavelengths. However, the UV spectra (Supplementary SI 4) revealed a change in the baseline with absorbances at $\approx$480 nm for the adrenochrome solutions and the 24 h

(−)-adrenaline in water solutions. These absorbances are responsible for the red colour produced and may indicate the presence of adrenochrome and other degradation products (Figure 2a). The red degraded solution (Figure 2c) is unsafe for patients and needs to be modified by excipients [32] to improve the stability (Supplementary SI 4 and SI 5).

However, it has been reported that adrenaline auto-injectors containing sodium metabisulfite can still degrade over time [32]. This is a consequence of the nucleophilic sulphite ions from the metabisulfite attacking adrenalines' alcohol groups [45] via mechanistic SN1 (nucleophilic substitution) at high pH or SN2 at low pH [32]. Further to this, racemisation can occur, forming sulfonate analogues (Figure 2a) but at a slower rate compared to adrenaline oxidation. The literature states that manufacturers compensate for this degradation by adding an additional 12% of the active isomer to formulations [45], but this induces variations in the formulation potency in regard to shelf life and usage.

### 3.2. Stress Tests

#### 3.2.1. Effects of Vibration

The vortex mixer has been successfully adapted to model onboard drone conditions of vibration where agitation speed and vibration frequency are approximately proportional [40]. The UV spectra (Figure 4) for the in vitro model formulations, regardless of vibration magnitude, overlapped perfectly and exhibited distinctive peaks at 280 and 221 nm. The CD spectra of the samples also overlapped perfectly and did not change significantly, indicating that (−)-adrenaline in this buffer system is stable. The intense negative ellipticity signal at 230 nm with the same magnitude was observed for all samples. The principal chromophore responsible for the UV and CD signals was the phenol chromophore $\pi \rightarrow \pi^*$ electron transitions. There was no positive ellipticity signal, emphasising that only the active isomer was present. No significant differences ($p > 0.05$) were observed between the samples tested at the three different speeds and the control samples.

Compared to the free base, the spectra for the ampoules and bitartrate formulations showed similar trends in UV and also similar but not identical trends in CD (SI 6). The absorption peaks at 280 and 221 nm were visible; however, there was a shift in CD signal from 230 to 216 nm and in intensity, due to the presence of the acid tartrate, which is a chiral molecule. There was no positive ellipticity signal, confirming that only the active isomer was present in both formulations. No significant differences were observed ($p > 0.05$) between the samples tested at 800 rpm and the control sample (Table 3).

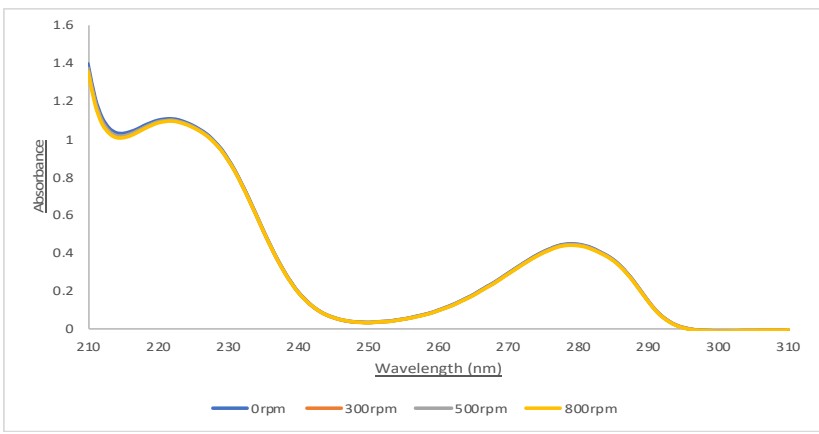

**Figure 4.** *Cont.*

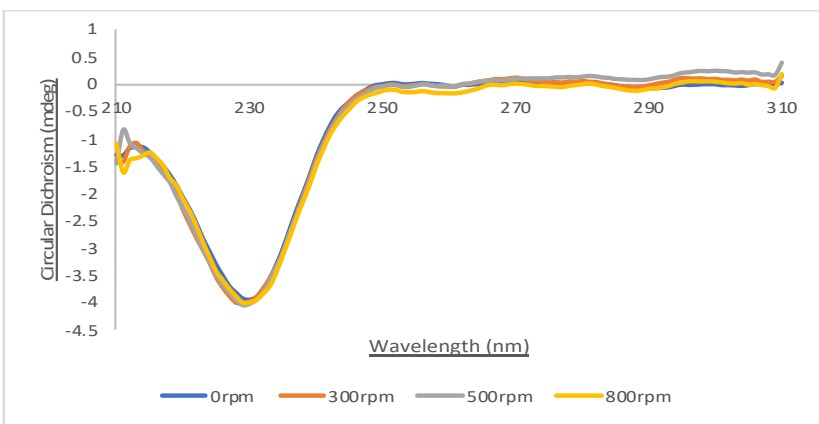

**Figure 4.** UV and CD spectra of in vitro formulations (0.5 mg/mL) after agitation for 30 min at 300, 500, and 800 rpm at 25 °C (*n* = 3).

**Table 3.** *p* values for the comparison of UV and CD spectra of in vitro (−)-adrenaline, bitartrate, and ampoule formulations (0.5 mg/mL) after vibration for 30 min at 300, 500, and 800 rpm at 25 °C with the control. Results were considered statistically different with the control when $p < 0.05$, as shown by (−)-adrenaline forced degradation in water after 24 h at 25 °C.

|  | Formulation | | | | | |
| --- | --- | --- | --- | --- | --- | --- |
|  | (−)-Adrenaline | | Bitartrate | | Ampoule | |
|  | CD | UV | CD | UV | CD | UV |
| **Vibration (rpm)** | | | | | | |
| 300 | 0.50 | 0.93 | - | - | - | - |
| 500 | 0.50 | 0.94 | - | - | - | - |
| 800 | 0.92 | 0.91 | 0.92 | 0.92 | 0.78 | 0.99 |
| Forced Degradation in Water | $5.47 \times 10^{-16}$ | $1.26 \times 10^{-8}$ | | | | |

These spectra indicate that vibrations that model take-off, flight, and landing, as reported by Hii et al. [40], do not cause changes in adrenaline conformation and concentration. However, more work is required to investigate how vibration is transferred from a drone's propellers and other moving parts via the transport container into the cargo's or medicine's packaging. Packaging configuration, packaging composition, the number and alignment of individual units will all influence the impact of vibration. The data shown in Table 3 indicate that in terms of vibration, the drone flight should have little effect on the chemical stability of adrenaline in an aqueous solution for injection, but a flight test is required to prove this prediction; see Section 3.3. The bitartrate and ampoule samples met the pharmacopeia visual and percentage label claim standards for adrenaline/epinephrine injections, but the in vitro samples, although clear and colourless, were above the BP percentage label claim. The increase in dose uniformity is most likely due to small differences in molar absorptivity of the free base compared to the salt forms of adrenaline, since there were no significant differences between the values obtained.

### 3.2.2. Effects of Temperature

The UV spectra at each temperature and time interval for the in vitro formulations (Supplementary SI 7) revealed distinctive peaks at 280 nm and 221 nm, which are indicative of the phenol chromophore electron transitions $\pi \rightarrow \pi^*$. There were no significant changes ($p > 0.05$) in UV for samples annealed between 4 and 40 °C over the 24 h period. However, the $\lambda_{max}$ and thus concentration for the formulation

at 65 °C after 3 h was much lower in comparison to the other spectra with a 75% dose uniformity (Figure 5), and as expected, it did not conform to BP standards.

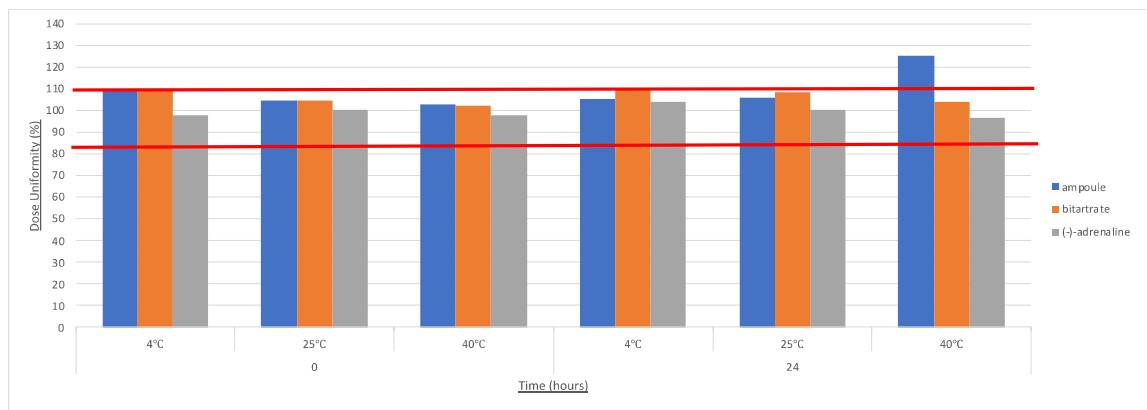

**Figure 5.** Adrenaline dose uniformity (%) of the in vitro (−)-adrenaline, bitartrate, and ampoule formulations after storage at 4, 25, and 40 °C for 0 and 24 h at 0.5 mg/mL (*n* = 3). The red lines indicate the BP percentage label claim upper and lower limits.

As temperature increased, there were slight changes in the intensity of the CD signal. The negative ellipticity signal at 230 nm was observed for all samples, confirming that there was no temperature-dependent racemisation.

The UV peaks at 280 nm and 221 nm for the ampoules and bitartrate formulations were visible. There were no significant differences in UV and CD (Table 4); however, the ampoule formulation at 40 °C after 24 h had a dose uniformity of 126% outside of the required percentage label claim. The variation in CD magnitude between samples were due to the different concentration of acid tartrate. There was no positive ellipticity signal, emphasising that the active isomer was only present in both formulations. All formulations remained clear and colourless, meeting the BP visual standards.

**Table 4.** *p* values for the comparison of UV and CD spectra of in vitro (−)-adrenaline, bitartrate, and ampoule formulations at 4, 25, 40, and 65 °C for 0, 0.5, 3, and 24 h at 0.5 mg/mL (*n* = 3), with the control. Results were considered statistically different when $p < 0.05$, as shown by (−)-adrenaline forced degradation in water after 24 h at 25 °C.

| | **Formulation** | | | | | | | | | |
| --- | --- | --- | --- | --- | --- | --- | --- | --- | --- | --- |
| | **(−)-Adrenaline** | | | | | | **Bitartrate** | | **Ampoule** | |
| | **CD** | | | **UV** | | | **CD** | **UV** | **CD** | **UV** |
| **Time (hours)** | 0.5 | 3 | 24 | 0.5 | 3 | 24 | 24 | 24 | 24 | 24 |
| **Temperature (°C)** | | | | | | | | | | |
| 4 | 0.45 | 0.06 | 0.22 | 0.72 | 0.99 | 0.99 | 0.64 | 0.92 | 0.89 | 0.88 |
| 25 | 0.70 | 0.98 | 0.81 | 0.98 | 0.94 | 0.98 | 0.59 | 0.91 | 0.79 | 0.93 |
| 40 | 0.02 | 0.94 | 0.23 | 0.98 | 0.96 | 0.99 | 0.90 | 0.90 | 0.35 | 0.16 |
| 65 | 0.36 | 0.91 | 0.97 | 0.87 | 0.45 | 0.77 | - | - | - | - |
| **Forced Degradation in Water** | | $5.47 \times 10^{-16}$ | | | | | $1.26 \times 10^{-8}$ | | | |

To check the conformation and the reversibility of the formulations, the spectra after heating and cooling were recovered to 25 °C. Reversibility and conformation recovery was apparent in all formulations, including the buffer solution at all temperatures and time intervals. This indicates that the formulation could be used by a patient after drone flight regardless of temperature changes and still have a high efficacy in any anaphylactic emergency. Comparable studies showed that adrenaline when stored at 2 °C and returned to 25 °C exhibited no statistically significant concentration changes, and after a 12 h freeze–thaw cycle, it still met BP standards [47]. Zenoi et al. also observed no statistically significant change in concentration when storing auto-injectors between 2 and 8 °C for

24 weeks [47], providing evidence that drone transportation is capable at such low temperatures and for long time periods.

Manufacturers provide specific storage requirements [27] for EpiPen®, warning patients to only store their device between 20 and 25 °C and not in the refrigerator [26], since temperature extremes promote product discolouration and dose discrepancies [28]. This is important when considering drone flight because as altitude increases, temperatures lower, and so onboard heating rather than cooling may be required to keep the EpiPen® at a stable temperature. In practice, the maximum height for flight is 400 ft; thus, the likelihood of extreme temperature differences due to altitude is low. However, transportation in colder climates could cause the auto-injector liquid, if not adequately insulated, to freeze; alternatively, it could cause cracks in the glass interior or malfunction of the injection mechanism—all must be investigated. Due to concerns with adrenaline quality and robustness at low temperatures, patient-friendly sublingual or inhaler forms could be an alternative for future consideration. However, the test data generated and the above literature [47] confirm that adrenaline quality in drone flight can be maintained at temperature ranges between 4 and 40 °C.

### 3.3. Drone Flight

Drone flights consisted of package loading, pre-flight checks, take-off, main flight phase, landing, and package unloading; this was repeated for three EpiPen® packages, A, B, and C, respectively. During flight, vibration was fairly stable and fell within the range observed by Hii et al. [40]. The maximum g-force was experienced during landing 2.45 g (±0.19), the minimum was experienced prior to take-off 1.23 g (±0.22), and during the main flight phase, the maximum was 1.82 g (±0.23). Temperature also remained stable during flight; however, thermograms identified variations in drone compartment temperature (Figure 6b). Changes due to heating via radiation from the sun depict that future drones will need to consider onboard heating or cooling devices to keep temperatures consistent and medication quality optimised.

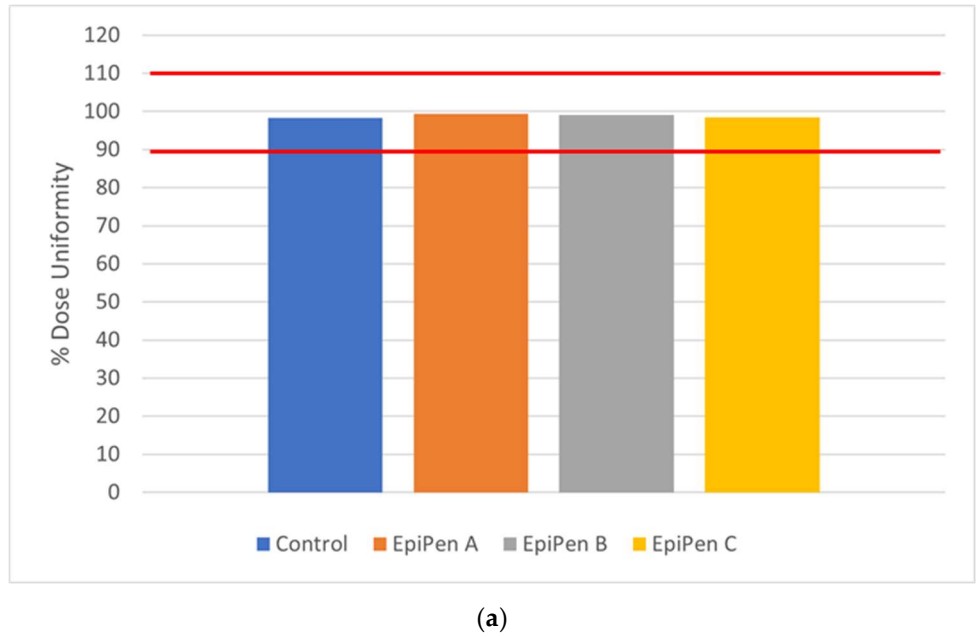

(**a**)

**Figure 6.** *Cont.*

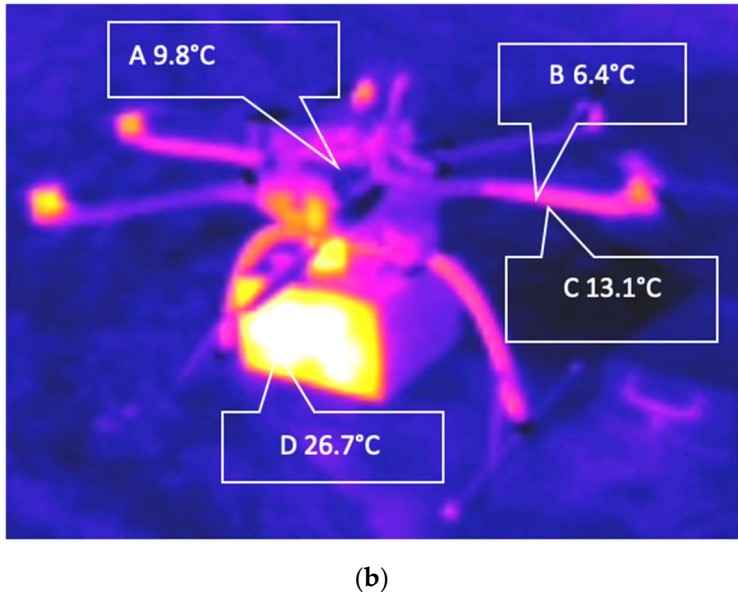

**(b)**

**Figure 6.** (**a**) Adrenaline dose uniformity (%) of EpiPen® after drone flight (*n* = 3, SD = ±0.49). The red lines indicate the BP percentage label claim upper and lower limits. (**b**) DJI Mavic 2 Enterprise Dual Drone Thermogram taken after flight identifying the variations in drone temperature on the transport box: A–D: Top, Propellers white surface, propeller black surface, back of the storage compartment.

After flight, EpiPens® were injected into sample vials for dilution, where no malfunction of the injection devices was observed. The UV peak at 280 nm and the negative ellipticity at 230 nm were visible in all samples (Table 5) with no significant differences observed (SI 8). The EpiPens® flown by drone conformed to BP standards in terms of concentration, visibility, and percentage dose uniformity: control (98.39%), A (99.44%), B (99.01%), and C (98.47%). This concluded that EpiPens® and adrenaline quality are not affected by drone delivery.

**Table 5.** *p* values for the comparison of UV and CD spectra of EpiPen® after drone flight of 10,732 m, at 12 m/s, 1.82 g (±0.23), 10.70 °C (±0.32), and 47.74% RH (±0.29), with the control. Results were considered statistically different when *p* < 0.05.

| | EpiPen® | | | | | |
| | EpiPen® A | | EpiPen® B | | EpiPen® C | |
| | CD | UV | CD | UV | CD | UV |
|---|---|---|---|---|---|---|
| *p* value | 0.55 | 0.97 | 0.11 | 0.98 | 0.73 | 0.99 |

It is crucial to highlight that the drone compartment was not bespoke; it was merely a "temporary transport device" perceived to be working beyond the edge of failure. However, despite this box, the EpiPen® quality surprisingly conformed to BP standards. Following this investigation, improvements have been made to provide a unique medical-specific payload box built for cold chain operations and dangerous goods, which can be attached to the drone (SI 9). Furthermore, the design of the box includes an outer heat reflective layer to maintain specific flight temperatures. Using this novel medical-specific payload box in future investigations would be far superior and beneficial.

Considering the drone transport of adrenaline EpiPens®, the payload box used in this study was fit for purpose even in a crash situation. As highlighted in the introduction, EpiPens® have passed rigorous drop tests, and the medicine regulators consider unintended use unlikely because of the clear instructions and warnings visible on the outer packaging. Furthermore, at the concentrations that the EpiPen delivers, adrenaline is not intrinsically dangerous. However, the novel medical-specific payload

box (SI 9) is required if the medical product to be transported is regarded as dangerous goods, e.g., cytotoxic drugs or biological samples. For such operations, carriage must align with the International Civil Aviation Organisation's (ICAO) technical instructions for the safe transport of dangerous goods by air. Following these instructions, the drone transportation of dangerous goods in the UK must have approval from the Civil Aviation Authority in conjunction with the appropriate operator training. For example, the CAA has approved training courses for the transport of biological samples by air, which permits operators to be approved for the transportation of UN3373 biological samples (category B) by drone. Such samples must be packaged in accordance with IATA Dangerous Goods Regulations, specifically packing instruction 650. PI 650 continues: "Packaging will be of sufficient strength and rigidity to withstand conditions associated with transport by drone. Several tiers of packaging will be used. Primary receptacles (e.g., blood vials) will be wrapped in an absorbent material and placed in a secondary leakproof packaging. This, in turn, will be wrapped in a cushioning material and secured in a rigid outer packaging. The outer packaging is dual purpose, providing thermal insulation and structural rigidity. Labels appropriate for the contents will be affixed to the outer package, which will then be inspected and stowed in the payload compartment of the drone in preparation for flight." Future operations will also need to take account of the pending European Union Aviation Safety Agency (EASA) regulations concerning crash-proof containers. Thus, future payload boxes must comply with PI 650, be able to withstand a crash situation, and ideally, they must also be tamper-proof.

For anaphylactic emergencies, future drones need to arrive at the scene within 5–10 min whilst adhering to the CAA and UK Drone Code. There needs to be technology implemented for communication with patients and to enhance the security of the supply chain, such as digital applications i.e., facial recognition or tamperproof boxes with personal keycodes, microphones, and cameras for visual assessments, since there is no medical professional at the scene. Additionally, lithium battery expense, battery life (due to rapid decreases during cold temperatures), payload, and achievable ranges must be addressed i.e., the use of hub-and-spoke locations or direct delivery from pharmacies/GP surgeries.

It is important to note that the Clogworks DarkMatter HX drone [48] can reach a maximum horizontal speed of 20 m/s and maximum vertical speed of 6 m/s; thus, our Billericay flight path could be used as a "typical" useful model for patients suffering from this disease state in the UK. There are more than 15,000 community pharmacies [49] and ≈9000 GP Surgeries spanning the UK, which are likely to provide achievable ranges and distances for the direct drone delivery of EpiPen® i.e., accessible within 10 km of housing. The majority of pharmacies and GP surgeries are located in areas of high population density and are more evenly distributed than ambulance services, enhancing the model. Thus, ≈80% of the UK would be sufficient for this model; however, timing and distance issues to reach patients could arise in more rural regions. The largest increase in anaphylactic emergencies over the past 5 years has occurred in London (167%), so it would be wise to initiate here first [20].

*3.4. Pharmacists' Perspectives Survey*

Fifty-five pharmacists participated in this study; 29 from the community, 19 from hospital, and seven from general practice. Overall, there was a positive reaction from pharmacists, with 73% agreeing that drones would be beneficial in delivering EpiPen® to patients in anaphylaxis emergencies. The pharmacists who approved of drone usage commented on the "potential catastrophic consequences" of anaphylaxis and how this "niche solution" is worth exploring. Many also provided answers associated with the following keywords: avoids traffic, cost-effective, and future delivery service. Analysis revealed that the younger generation of pharmacists (55%) seemed to be more accepting, aware, and had confidence in the concept of drone delivery.

In accordance with global auto-injector research [30], all pharmacists (100%) revealed that they were currently experiencing major logistical challenges in terms of delivery and problems with auto-injector availability due to national shortages and various recalls of all licensed adrenaline auto-injector brands in the UK. As a result, this is causing major issues to their patients who are

at risk of anaphylaxis, especially those with a severe risk who must carry two auto-injectors at all times [23], since failure in positive response to the initial injection may occur, as a consequence of advancing anaphylaxis or delayed, inadequate dosage—this is fatal [17]. It is reasonable to suggest that shortages will mean that there are patients carrying just one or even no auto-injectors. It was noted that supply interruptions have caused manufacturing stability data to be reviewed, where certain brands are advising new extended shelf lives [50]. Supply chain issues have also raised awareness of other available options of adrenaline, i.e., pre-filled syringes (which are often not considered and are of least preference due to their complex application technique) [21].

There were mixed reviews in terms of patient acceptance—34% of pharmacists felt that their patients would be neutral to drone delivery, 40% were accepting and 26% were unaccepting. Thirty-eight percent of pharmacists had low confidence in adrenaline quality being maintained during flight, and 44% of pharmacists were "somewhat confident" in EpiPen® robustness and commented on the specific storage recommendations. Pharmacists also had low confidence in supply chain security (76%) and whether drones could deliver EpiPen® within an achievable range (44%) (Figure 7). Concerns with drone usage addressed the possibility of serious injuries, i.e., finger laceration arising from rapidly rotating propellers, drone speed, needle safety, drone failure, and security. Specific issues isolated were disapproval of EpiPen® drone delivery, which was based on "negative public perceptions, "snooping", and privacy issues in densely populated areas". Furthermore, there was concern whether the NHS would "invest a large sum of money and time into a drone service".

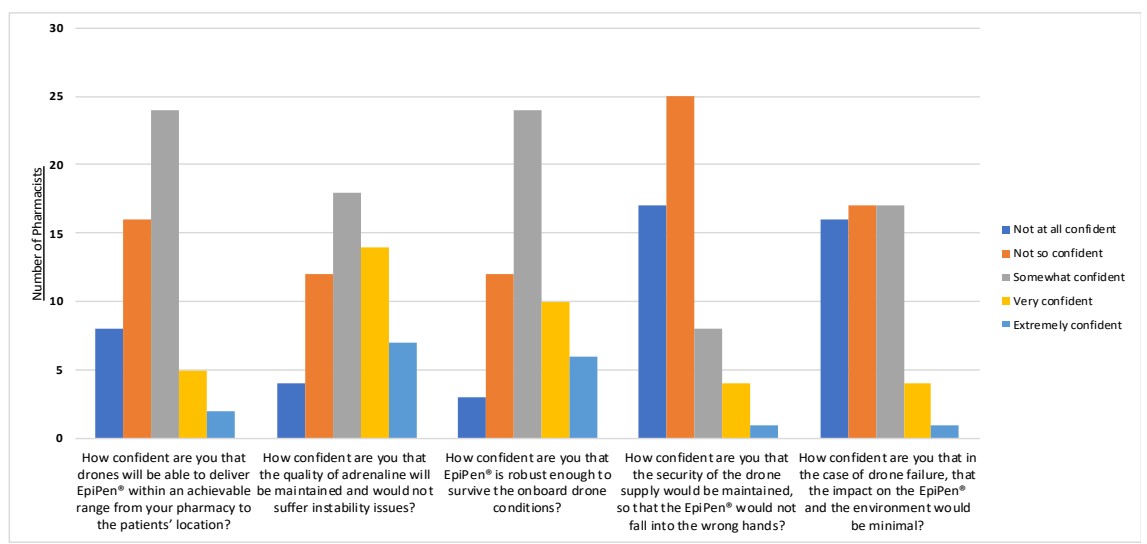

**Figure 7.** Pharmacists confidence in the drone delivery of medicines (*n* = 55). Five questions were devised to assess pharmacist's confidence in the possible range, quality and robustness, security, and safety associated with drone delivery of EpiPen®.

Although the survey showed that pharmacists' perceived drones as a good concept, our survey revealed that pharmacists still had low confidence in the possible range, quality, security, and safety associated with the drone delivery of medicines, and ultimately, the effectiveness of drone delivery may depend on the patient's ability to self-administer their medicine, especially if it is in the form of an injection. However, the experimental data obtained in this investigation from drone flight tests addresses many of the issues raised, and the findings this study should be circulated to increase pharmacists' confidence in drone delivery. For full details of the survey, see Supplementary SI 1.

## 4. Conclusions

The opportunities for drones in healthcare are extensive and rewarding. The COVID-19 pandemic has sparked a rapid amplification of healthcare drones across the globe. The proliferation in commercial

collaborations to increase on-demand medical deliveries has kept those most vulnerable within society functional. The attributes of high speed, precision, and flexibility make drones particularly attractive for emergency healthcare missions. This study provides essential stability data confirming the potential for drones to provide a viable EpiPen® delivery service during an anaphylactic episode. This paper established that there were no significant changes in adrenaline concentration and chiral composition during drone transportation and under simulated drone conditions—with formulations conforming reproducibly to the pharmacopeia standards. Additionally, the practical data obtained address the concerns raised by pharmacists, and its analysis should be circulated. With the acceleration in the advancement of drone technology and robotic systems, there is no doubt that the use of drones or similar devices will become more prevalent in the future. However, there is a necessity for all clinical professionals to be educated about drones to alleviate any concerns. Further to this, there is the requirement of regulatory acceptance from the CAA before the implementation, integration, and optimisation of drone technology into future healthcare can occur.

This research should serve as a framework for future healthcare drone delivery investigations. The authors recommend that when designing future missions, one must conduct a perceptions survey on the relevant group of medical professionals, because their insights, acceptance, and concerns are extremely valuable for the design and evaluation of the mission. Additionally, one must consider the medication quality and safety parameters before, during, and after flight (i.e., meeting the pharmacopeia recommendations), and most importantly, one must also consider patient safety.

**Supplementary Materials:** The following are available online at http://www.mdpi.com/2504-446X/4/4/66/s1, Figure S1: Pharmacists Perceptions (Survey & Results), Figure S2: Buffer Preparation and Model Formulations, Figure S3: Skyport Information, Figure S4: Stability Investigation, Figure S5: The Effects of Light, Figure S6: The Effects of Vibration, Figure S7: The Effect of Temperature, Figure S8: Drone Flight, Figure S9: Medical-Specific Payload Box.

**Author Contributions:** Conceptualization, S.B., P.C. & P.G.R.; Methodology, S.B., T.T.B., A.D., P.C., A.B., J.G., & P.G.R.; Investigation & Results, S.B., T.T.B., J.G., & P.G.R.; Data work up & curation S.B. & T.T.B.; Analysis & Interpretation, S.B., A.D., P.C. & P.G.R.; Resources, T.T.B., A.D., A.B. & P.G.R.; Writing—original draft preparation, S.B.; Writing—review & editing, S.B., T.T.B., A.D., P.C., A.B. & P.G.R.; Supervision, P.C. & P.G.R. All authors have read and agreed to the published version of the manuscript.

**Funding:** This research received no external funding.

**Acknowledgments:** The authors would like to thank the Centre for Biomolecular Spectroscopy funded by a capital award by the Wellcome Trust; Fitzroy Calliste for sourcing EpiPen®; Alastair Skitmore for advice concerning permissions for operations and packing instructions and the Skyports Team, for allowing the KCL Drone Team to use their facilities.

**Conflicts of Interest:** The authors declare no conflict of interest.

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
