# Peer review of "An Evaluation of the Drone Delivery of Adrenaline Auto-Injectors for Anaphylaxis: Pharmacists’ Perceptions, Acceptance, and Concerns"

_drones, doi:10.3390/drones4040066_

Round 1

Reviewer 1 Report

This is a generally well written paper investigating the potential benefits, risks and impacts of drone transportation on the stability of adrenaline. Original primary research has been undertaken in terms of simulation, live trials and consultation with pharmacists. Some comments are provided below which the authors should consider when revising their paper. This is a genuinely novel contribution to the field and should be published.

1)      ‘Anaphylaxis is a rapid and serious systemic allergic reaction with a lifetime prevalence of 2% 85 [17], commonly triggered by food [18], medications or insect stings [19].’ …. Are certain people pre-disposed to it and therefore would be expected to carry an EpiPen with them?

2)      I think the general description of adrenaline needs to cover its goods classification (presumably adrenergic bronchodilators are classed as dangerous goods from a transportation perspective and would require special packaging (3-layer as in PI650) to be transported by land vehicles). What are the implications of being transported in a UAV from a safety perspective to those on the ground and what are the key risks to the product of UAV flight? This warrants some basic discussion from the outset to get across the core issues and help set the research questions. Does adrenalin come under UN3373?

3)      ‘A vortex mixer (ZX3, VELP) fabricated the range of vibrations that the EpiPen® would experience during drone take-off, flight, and landing’ …. Some base description of these forces is warranted and how they differ with different type of UAV. What has been observed from the literature?

4)      Different temperatures were experimented with. What temperature should EpiPen’s ideally be transported at?

5)      Section 3 would be better starting with the stability analysis and stress tests as they have been the major focus of the preceeding section. The views from the pharmacists survey can be used in the final discussion and conclusions section.

6)      Section 3.2, ‘stability investigation’ is not easy to decipher for a reader not rehearsed in medical practice. The key thing that is missing from this section is the relevance of the findings from the drone operations perspective. What are the key issues in terms of the stability of adrenaline when in the flight stage?

7)      ‘Vibration during take-off, flight and landing does not cause changes in adrenaline conformation and concentration’.  This is a useful finding but there is no commentary on how vibration transferred through to the cargo via the packaging. Would different packaging configurations from different packaging manufacturers have different results? The whole flight testing needs more detail adding to describe what happened, how many flights, what was the flight phasing etc.

8)      ‘Improvements have been made to provide a unique medical-specific payload box, built for cold chain operations which can be attached to the drone (SI 9)’. It would be interesting to know if this cargo container is designed with the carriage of dangerous goods in mind and whether it conforms to the pending EASA regs on crash proof containers for drones. Some discussion here is warranted in relation to the containment of the adrenalin in a crash situation and the likely consequences of unintended gestation.  

Author Response

The authors really appreciate the positive and supportive comments from reviewer 1. The minor corrections suggested by the reviewer have all been addressed in the revised manuscript, and as a result the paper has been considerably enhanced.

1)      ‘Anaphylaxis is a rapid and serious systemic allergic reaction with a lifetime prevalence of 2% 85 [17], commonly triggered by food [18], medications or insect stings [19].’ …. Are certain people pre-disposed to it and therefore would be expected to carry an EpiPen with them?

Response: An extra section has been added to give more of the background of EpiPen use. This begins on line 98 and is also given below;

NHS clinical guidelines recommend that those with potentially serious allergies should carry two in-date adrenaline auto-injectors as an emergency treatment for anaphylaxis. Patients who have allergies towards insect stings, peanuts, milk, and seafood fall within this group, furthermore people with asthma and the allergic skin condition atopic eczema have a greater risk of developing anaphylaxis. The National Institute for health and Care Excellence, NICE, guidance states that it is important to use an adrenaline auto‑injector as soon as possible if an anaphylactic reaction is suspected.

2)      I think the general description of adrenaline needs to cover its goods classification (presumably adrenergic bronchodilators are classed as dangerous goods from a transportation perspective and would require special packaging (3-layer as in PI650) to be transported by land vehicles). What are the implications of being transported in a UAV from a safety perspective to those on the ground and what are the key risks to the product of UAV flight? This warrants some basic discussion from the outset to get across the core issues and help set the research questions. Does adrenalin come under UN3373?

Response: The goods classification has been extensively covered in the new section added at line 144 in the revised manuscript and this section is given below. Apologies this has turned out to be quite a long addition, however adrenaline makes an excellent example of the issues concerning the air transport of a medicine. The pure drug substance (adrenaline) is classed as dangerous goods but the drug product (the medicine, i.e. EpiPen) is not classed as a dangerous within the regulations, this has been confirmed by the manufacturer. For the readership of the Journal Drones this observation is probably quite surprising and has not been clearly highlighted in the UAV literature before, thus in order to include all of the relevant regulations, governing bodies and to clearly make an evidence-based case, this section required extensive discussion. The authors feel that the information included in the new section is all relevant.

The goods classifications for pure adrenaline (drug substance) and when formulated into an injection for an EpiPen® (drug product) are quite different in terms of the transport perspective. Adrenaline (epinephrine) in its solid acid tartrate form is classified, according to the British Pharmacopoeia safety data sheet (1907/2006/EC, Article 31), as a toxic solid, UN2811, packing group III (presenting low danger) and has an International Air Transport Associated (IATA) classification of 6.1. This means that the transportation of pure adrenaline by air requires performance-oriented packaging that must have enough strength to withstand shocks, loadings and the typical changes in atmospheric pressure that occur during flight. This would normally be achieved by a primary container, usually plastic, for example high density polyethylene, which is inert and able to avoid leakage at an internal pressure of 95 kPa. The primary package would be placed within rigid outer packaging but separated by a cushioning material, e.g. polystyrene or engineered cardboard. A clear label would be required, following UN guidelines indicating toxic material and the IATA hazard class of 6.1. Transport of 5 kg is permissible on a passenger aircraft and 50kg is the limit for transportation on a cargo aircraft. Many medicines, especially EpiPen® adrenaline auto-injectors, are not treated as dangerous goods. Medicines that are dangerous for carriage are those which contain flammable or volatile ingredients, for example the propellants used in a pressurized metered dose inhalers or are particularly toxic, for example chemotherapies and the cytotoxic drugs used in the treatment of cancer, these are covered by UN numbers 1851, 3248 and 3249. These numbers are used to identify hazardous substances within international transport systems and are assigned by the United Nations Committee of Experts on the Transport of Dangerous Goods.

The rationale for not subjecting a particular medicine to IATA transport regulations is frequently based on the small amount of the active pharmaceutical ingredient (or drug) present, typically in the milligram range, which is far below the UK’s Health & Safety Executive’s limited quantity exemption for the international carriage of dangerous goods, and the acknowledgement that the medicine is contained within packaging already approved for retail sale and distribution for personal and household use (UN – SP601).  In the UK & Europe packaging approval is governed by the Medicines & Healthcare products Regulatory Agency and the European Medicines Agency respectively, within their Good Distribution Practice guidance. Similar regulations are applied worldwide, e.g. by the FDA in the US. Thus, solid adrenaline falls within the regulations concerning the air transportation of dangerous goods, but adrenaline autoinjectors do not. For example, the materials safety data sheet for adrenaline (epinephrine) injection (1mg/mL) published by a manufacturer of the drug Hospira & Pfizer, , states that the injection is not regulated for transport under USDOT, EUADR, IATA or IMDG regulations. (US Department of Transport, USDOT; European Agreement concerning the International Carriage of Dangerous Goods by Road, EUADR; International Maritime Dangerous Goods code, IMDG). Therefore, the authors’ choice of EpiPen® adrenaline auto injectors allows the evaluation of drone delivery without the requirement for extra packaging, and keeps the studies’ focus on the key research questions, i.e. the impact of drone flight on adrenaline stability and the perceptions of pharmacists towards drone deliveries. It should be noted that CAA approval is required for all commercial operations, and medical deliveries are no exception.  Many medical cargos of interest are classified as dangerous goods, for example samples used in blood tests (covered by UN3373), thus robust safety cases will be required to gain regulatory approval for all future UAV commercial medical deliveries.

EpiPen® autoinjectors have successfully passed drop tests and load challenge, without any loss of performance characteristics i.e. after the mechanical stress EpiPen® was able to consistently deliver the correct volume dose but the chemical stability of adrenaline was not measured [37]. Mechanical robustness is not surprising as EpiPen®s are designed in rigid packaging to allow them to be carried in bags & coats with limited risk. Therefore, the specific safety perspectives associated with transport of EpiPen®s by drone are adrenaline stability during flight, and unexpected or unplanned landing. In the case of unplanned landing, tampering and diversion need to be considered, as unlike road carriage, no operator would be in the immediate vicinity to protect the cargo. The outside of EpiPen® packaging has clear safety warnings, and there is little financial benefit in the UK associated with diversion (unregulated re-sale) because of the national prescription prepayment scheme. This may not be the case internationally, for example EpiPen® costs approximately $400 in the US, which would warrant the development of extra protection. Anti-tampering measures would also reduce the risk of bystanders harming themselves if they are unfamiliar with EpiPen®, such themes are taken up at the end of this paper. 

Reference added:

  1. Schwirtz, A. and Seeger, H., 2012. Comparison of the robustness and functionality of three adrenaline auto-injectors. Journal of asthma and allergy5, p.39

3)      ‘A vortex mixer (ZX3, VELP) fabricated the range of vibrations that the EpiPen® would experience during drone take-off, flight, and landing’ …. Some base description of these forces is warranted and how they differ with different type of UAV. What has been observed from the literature?

Response: The following section has been added at line 247 to address the reviewer’s comment.

Hii et al. correlated vortex mixer speed to vibrational frequency using vibration data loggers and mapped these using the same loggers to the vibration and acceleration g-forces experienced during multirotor drone flight [39]. During vortex mixing an off-centered rubber cup oscillates in a circular motion, the sample vial is placed into this cup and the oscillatory motion is transmitted to the liquid inside creating a vortex. This intense shaking and turbulent flow induce shear forces and cavitation within the solution. Typical rotational speeds for the propellers of small multirotor drones are between 4,000 and 6000 rpm, typically near 5000 rpm in flight [40].  This is much greater than the vortex mixer can achieve. However, the symmetrical alignment of rotors in such drones is designed to reduce vibration and only in lift off and take off are significant vibrations observed, for Hii et al. this was 3.4Hz, with an upper limit peak of 10Hz. During normal horizontal flight this dropped to only 0.1Hz. However, when rotors are damaged, or poorly balanced, vibration will increase with a concomitant lowering of thrust performance, [41]. The g-force also needs to be considered so in the present study this was measured directly on board the flight, see section 2.5 and Figure 1d, and the stability of the transported adrenaline measured post flight, after exposure to this force.

References added:

  1. Deters, R.W., Kleinke, S. and Selig, M.S., 2017. Static testing of propulsion elements for small multirotor unmanned aerial vehicles. In 35th AIAA Applied Aerodynamics Conference(p. 3743).

  1. Kuantama, E., Moldovan, O.G., Ţarcă, I., Vesselényi, T. and Ţarcă, R., 2019. Analysis of quadcopter propeller vibration based on laser vibrometer. Journal of Low Frequency Noise, Vibration and Active Control, doi: 10.1177/1461348419866292.

4)      Different temperatures were experimented with. What temperature should EpiPen’s ideally be transported at?

Response: The comment has been addressed by the addition of the section beginning on line 269 of the revised manuscript

EpiPen® adrenaline autoinjector manufacturers and the MHRA recommend storage and transport at temperatures are between 20 and 25°C but temperature excursions between 15 and 30°C are permitted.

5)      Section 3 would be better starting with the stability analysis and stress tests as they have been the major focus of the preceeding section. The views from the pharmacists survey can be used in the final discussion and conclusions section.

Response: This been carried out in the revised manuscript.

6)      Section 3.2, ‘stability investigation’ is not easy to decipher for a reader not rehearsed in medical practice. The key thing that is missing from this section is the relevance of the findings from the drone operations perspective. What are the key issues in terms of the stability of adrenaline when in the flight stage?

Response: The reviewer makes an excellent point, thus a little more detail has been given concerning the rationale of the stability investigation and how it fits within the medicine regulatory landscape. This is covered in the new section which begins on line 337 of the revised manuscript and is also given below.

ICH represents the International Council for Harmonisation of the technical requirements for pharmaceuticals for human use, and Q1A (R2) guidelines are associated with stability testing requirements. The purpose of such stability testing is to provide evidence on how the quality of a drug or medicine varies with time under the influence of environmental factors, for example temperature and humidity. For EpiPen® Adrenaline Autoinjectors such testing has already been carried out, and quality indicating analytical procedures have allowed the recommend storage conditions and shelf life to be set. The approach here was to use a UV (ultraviolet) and CD spectroscopy assays to detect changes in concentration and chiral composition resulting from a loss of stability and the generation of well documented degradation products, see figures 2 & 3. Thus, quite simply if the adrenaline solution changed colour or became cloudy, or the UV or CD spectra differed from the control, then it would be concluded that it was unstable towards the particular environmental challenge investigated. Likewise, the absence of colour change and no significance difference in both the UV and CD spectra indicated stability within the experimental time frame. EpiPens are enclosed units, so humidity was not considered, but temperature and vibration were considered as environmental factors. Vibration originating from drone flight has been addressed in section 2.4.1. Increase in load mass reduces the range of a drone flight [43], so here the flight stages were modelled without insulation or temperature control, as both would add mass. WHO has designated 4 climatic zones, temperate, subtropical, hot & dry and hot & humid, the maximum accelerated stability testing temperature representing hot climates is 40°C, the highest recorded temperatures are in the low 50’s and thus to explore the edge of failure, a 65°C temperature limit was investigated here. Both WHO and the ICH recommend stability testing at normal refrigerator temperatures and so 4°C was chosen as the lower limit. Our study has been designed to conform to world-wide aviation regulations and so flight altitudes would be within 140ft and so ground temperatures may be extrapolated to this relatively low height. Drone operations are occurring in all WHO designated climatic zones, and thus the stability environments investigated are well within the temperatures potentially encountered during loading, lift-off, flight, and the landing stages of future drone operations.

7)      ‘Vibration during take-off, flight and landing does not cause changes in adrenaline conformation and concentration’.  This is a useful finding but there is no commentary on how vibration transferred through to the cargo via the packaging. Would different packaging configurations from different packaging manufacturers have different results? The whole flight testing needs more detail adding to describe what happened, how many flights, what was the flight phasing etc.

Response: Configuration was not investigated and to address the reviewers point concerning how vibration is transferred, the following has been added in mitigation, incorporating edits on line 451, and new sections beginning on line 452, and 516 – which provides more detail of the flight testing. Additions are given below.

Line 452: These spectra indicate that vibrations which model take-off, flight and landing, as reported by Hii et al. [39], do not cause changes in adrenaline conformation and concentration. However, more work is required to investigate how vibration is transferred from a drone’s propellers and other moving parts via the transport container into the cargo’s or medicine’s packaging. Packaging configuration, packaging composition, the number and alignment of individual units, will all influence the impact of vibration. The data shown in table 3 indicates that in terms of vibration, the drone flight should have little effect on the chemical stability of adrenaline in an aqueous solution for injection, but a flight test is required to prove this prediction, see section 3.4.

Line 516:  Drone flights consisted of package loading, pre-flight checks, takeoff, main flight phase, landing, package unloading, this was repeated for three EpiPen® packages, A, B & C, respectively. During flight vibration was fairly stable and fell within the range observed by Hii et al. [39]. The maximum g-force experienced during landing 2.45g (±0.19), minimum prior to take-off 1.23g (±0.22) and during the main flight phase the maximum was 1.82g (±0.23).

8)      ‘Improvements have been made to provide a unique medical-specific payload box, built for cold chain operations which can be attached to the drone (SI 9)’. It would be interesting to know if this cargo container is designed with the carriage of dangerous goods in mind and whether it conforms to the pending EASA regs on crash proof containers for drones. Some discussion here is warranted in relation to the containment of the adrenalin in a crash situation and the likely consequences of unintended gestation.

Response: Again reviewer 1 makes an excellent and very useful point to improve the quality of the discussion. As a result, the authors have added an in-depth discussion of the packaging requirements for the drone delivery of dangerous goods, given in a new section within the revised manuscript beginning on line 541. But also to address the reviewer’s opening remark an addition has been made on line 538, these are shown in read in the revised manuscript, but the section from line 534 is given below for clarity.

Line 534: It is crucial to highlight that the drone compartment was not bespoke, merely a ‘temporary transport device’ perceived to be working beyond the edge of failure. However, despite this box, the EpiPen® quality surprisingly conformed to BP standards. Following this investigation, improvements have been made to provide a unique medical-specific payload box, built for cold chain operations and dangerous goods which can be attached to the drone (SI 9). Further, the design of the box includes an outer heat reflective layer to maintain specific flight temperatures. Using this novel medical-specific payload box in future investigations would be far superior and beneficial.

Considering the drone transport of adrenaline EpiPen®s, the payload box used in this study was fit for purpose even in a crash situation. As highlighted in the introduction, EpiPen®s have passed rigorous drop tests and the medicines regulators considers unintended use unlikely because of the clear instructions and warnings visible on the outer packaging. Furthermore, at the concentrations that the EpiPen delivers, adrenaline is not intrinsically dangerous. The novel medical specific payload box (SI 9) is required, however, if the medical product to be transported is regarded as dangerous goods, e.g. cytotoxic drugs, or biological samples. For such operations, carriage must align with the International Civil Aviation Organisation’s (ICAO) technical instructions for the safe transport of dangerous goods by air. Following these instructions, drone transportation of dangerous goods in the UK must have approval from the Civil Aviation Authority, in conjunction with the appropriate operator training. For example, the CAA has approved training courses for the transport of biological samples by air, which permits operators to be approved for the transportation of UN3373 biological samples (category B) by drone. Such samples must be packaged in accordance with IATA Dangerous Goods Regulations specifically packing instruction 650. PI 650 continues: “Packaging will be of sufficient strength and rigidity to withstand conditions associated with transport by drone. Several tiers of packaging will be used. Primary receptacles (e.g. blood vials) will be wrapped in an absorbent material and placed in a secondary leakproof packaging. This, in-turn, will be wrapped in a cushioning material and secured in a rigid outer packaging. The outer packaging is dual purpose, providing thermal insulation and structural rigidity. Labels appropriate for the contents will be affixed to the outer package, which will then be inspected and stowed in the payload compartment of the drone in preparation for flight.” Future operations will also need to take account of the pending European Union Aviation Safety Agency (EASA) regulations concerning crash-proof containers. Thus, future payload boxes must comply with PI 650, be able to withstand a crash situation and ideally must also be tamper-proof.

Reviewer 2 Report

UAV is typically defined as Unmanned Aerial Vehicle. 

The route and flight of the drone should be explained in more detail. An overhead map detailing the route should be included in Figure 1. 

Author Response

The authors appreciate the positive response reviewer 2 had towards the submitted manuscript. Their minor revision is given below, with the authors response following it.

 1) The route and flight of the drone should be explained in more detail. An overhead map detailing the route should be included in Figure 1. 

Response: The reviewer makes an excellent point, and the suggested addition of a detailed map has been made to figure 1. More information about safety, permissions for operations and the flight details have been added in the methods section, the revision begins on line 299 and is also given below.

Line 299: Flight protocols implemented by the CAA were followed. Specifically, the drone flights where conducted under Skyports’ permission for operations and strictly adhering to their approved safety protocols at all times. Skyports is approved by the Civil Aviation Authority to operate within visual line of sight (VLOS) and within extended visual line of sight (EVLOS) anywhere in the UK. Additionally, Skyports has permission to operate Beyond Visual Line of Sight (BVLOS) in certain locations. The flights at Billericay were conducted under Skyports EVLOS regulatory approvals. This permission granted by the CAA includes a full Operating Safety Case (OSC) prepared by Skyports which extensively details the operation, processes, safety procedures and emergency procedures which comprise a drone delivery operation.